# Multimodal Imaging and Phototherapy of Cancer and Bacterial Infection by Graphene and Related Nanocomposites

**DOI:** 10.3390/molecules27175588

**Published:** 2022-08-30

**Authors:** Ganesh Gollavelli, Anil V. Ghule, Yong-Chien Ling

**Affiliations:** 1Department of Humanities and Basic Sciences, Aditya Engineering College, Surampalem, Jawaharlal Nehru Technological University Kakinada, Kakinada 533437, Andhra Pradesh, India; 2Department of Chemistry, Shivaji University, Kolhapur 416004, Maharashtra, India; 3Department of Chemistry, National Tsing Hua University, Hsinchu 30013, Taiwan

**Keywords:** graphene, nanocomposites, multimodal imaging, phototherapy, theranostics, cancer, bacterial infection

## Abstract

The advancements in nanotechnology and nanomedicine are projected to solve many glitches in medicine, especially in the fields of cancer and infectious diseases, which are ranked in the top five most dangerous deadly diseases worldwide by the WHO. There is great concern to eradicate these problems with accurate diagnosis and therapies. Among many developed therapeutic models, near infra-red mediated phototherapy is a non-invasive technique used to invade many persistent tumors and bacterial infections with less inflammation compared with traditional therapeutic models such as radiation therapy, chemotherapy, and surgeries. Herein, we firstly summarize the up-to-date research on graphene phototheranostics for a better understanding of this field of research. We discuss the preparation and functionalization of graphene nanomaterials with various biocompatible components, such as metals, metal oxides, polymers, photosensitizers, and drugs, through covalent and noncovalent approaches. The multifunctional nanographene is used to diagnose the disease with confocal laser scanning microscopy, magnetic resonance imaging computed tomography, positron emission tomography, photoacoustic imaging, Raman, and ToF-SMIS to visualize inside the biological system for imaging-guided therapy are discussed. Further, treatment of disease by photothermal and photodynamic therapies against different cancers and bacterial infections are carefully conferred herein along with challenges and future perspectives.

## 1. Introduction

Humankind have faced many threats, especially from cancer and infectious diseases, in the past and in the current times. These problems have remained persistent for many decades. Science has provided remedies alongside many religious beliefs, especially during the pandemic times. This scenario increased the need for non-invasive, economic, therapeutic models to fight cancer, Alzheimer’s disease, cardiovascular disease, influenza, COVID-19, and other microbial infections, and existing diseases [1,2,3,4]. Scientific advancements are required to find solutions to these problems. Innovations in science have provided many therapeutic models, such as chemotherapy and surgeries, after traditional treatment methods such as Chinese medicine and Indian. Innovations in nanotechnology and nanomedicine aim to provide better solutions in medicine [5,6,7,8]. Nanotechnology offers small size delivery systems inside cellular and subcellular levels owing to high surface area to carry many therapeutic drugs with biocompatibility and inherent theranostic properties [9].

Theranostics is an emerging field in nanomedicine which may provide simple, economic diagnoses and therapy solutions to many cancers and infectious diseases. Rather than rely on single diagnosis and therapy models, multiple practices are important to provide accurate results of disease confirmation and cure. Nanomaterials with multiple diagnosis and therapeutic characteristics are highly desired in nanomedicine [10,11]. The current diagnosis techniques for cancer and infectious diseases in research are Confocal Laser Scanning Microscopy (CLSM), Magnetic Resonance imaging (MRI), Computed Tomography (CT), Positron Emission Tomography (PET), Raman, and Time-of-Flight Secondary Ion Mass Spectrometry (ToF-SIMS). However, each model has its own advantages and disadvantage [12,13]. Other than multiple imaging guided techniques, multiple therapeutic models are also important, and chemotherapy, immunotherapy, gene therapy, and surgeries which can provide good results [14,15,16,17]. However, these treatments may prone to some kind of tissue damage and inevitable side effects [18,19,20,21].

In recent years, phototherapy has become emerging research topic in nanomedicine to treat cancer and bacterial infections [22]. Phototherapy is a non-invasive technique due to its usage of low laser powers and short time interactions to the patent [23]. This is due to the utilization of low energy NIR light which has better tissue penetration in biological systems than visible and UV light, which may burn the skin and harm the patient [24]. Any system which can absorb NIR light and create a local heat to burn tumors and bacterial cells would be beneficial to nanomedicine [25]. Many nanomaterials with different size, shape, and biofunctionality have been demonstrated to target cancer and bacterial invasion [26,27,28]. The most successful photo and chemotherapeutic nanomaterials, such as Au, Ag, Fe, carbon, and polymeric nanomaterials, are well studied [29]. Due to its very good biocompatibility, low toxicity, tunable size, and high surface areas, we selected 2D graphene and reviewed the status quo of this nanomaterial in nanomedicine and theranostics [30].

Graphene is an allotropic form of carbon where the carbons are arranged in a 2D hexagonal chicken-net-like network which can offer high surface area, better electrical and thermal conductivity with optical transparency, and tuneable surface functionality with the olefin carbon network [31,32]. The intriguing properties of size, shape, and toxicity of graphene and graphene-related nanomaterials, such as graphene oxide (GO), reduced graphene oxide (RGO), and functionalized graphene nanocomposites (GNCs), are investigated in this review for multimodal imaging guided targeted phototherapy. Herein, we discuss the preparation of GNCs functionalization with many metals, metal oxides, polymers, photsensitizers, as well as other therapeutic drugs by covalent and non-covalent approaches to treat malignant tumors and antibiotic resistant bacterial infections by NIR triggered photothermal therapy (PTT) and photodynamic therapy (PDT) as well as synergistic effects of other combination therapies (Figure 1).

## 2. Preparation of Graphene Nanocomposites

### 2.1. Graphene Oxide

GO belongs to the graphene family which is a densely packed honeycomb-like structure made from a sheet of sp^2^ and sp^3^ bonded carbon atoms. Graphene nanomaterials have benefits such as high mechanical strength, Young’s modulus, surface area, conductivity, and carrier mobility, making them a perfect nanomaterial for various applications [33]. Graphene and its derivatives are widely used due to their excellent inherent properties and extraordinary composition in drug delivery, cancer treatment, biosensing, and bioimaging. Apart from these advantages of GO, the study has also focused on its toxicity and demonstrates GNCs are less toxic than carbon nanotubes. This outcome supports the use of GNCs for cancer and hyperthermia treatment [34]. Figure 2 presents the various types of graphene and their composites, preparation, and biological applications. Types, preparation, properties, functionalization, and focused therapeutic applications of graphene nanomaterials are also shown in Figure 2. Researchers are currently giving particular attention to the preparation of single-layered GO from graphite, by using strong oxidizing agents and concentrated acids, because of its extensive applications in the biomedical field [35]. GO contains epoxy, carboxyl, carbonyl, and hydroxyl functional groups that make it hydrophilic and biocompatible [33].

The graphene was synthesized by various methods in which the top–down and bottom–up approaches are generally used. In the top–down method, discrete graphene sheets are synthesized by breaking a stacked layer of graphite. The top–down approach includes micromechanical cleavage, thermal reduction, and electrochemical exfoliation whereas, chemical vapor deposition is included in the bottom–up approach [35]. Among the several preparation methods of graphene, the reduction of GO has gained significant attention because of its low-cost, ease of implementation, as well as variety of reducing agents and synthesis procedures [36]. Moreover, for the preparation of GO the Staudenmaier method, Brodie method, Hummers’ method and their modified versions are well known and widely used [37]. However, the Hummers’ method showed the degree of oxidation to be more compared with the other methods [38]. In brief, in the Hummers’ method the graphite flakes were mixed with H_2_SO_4_ and NaNO_3_ solution under an ice bath. Then, KMnO_4_ was added to the above mixture with constant stirring. Due to the addition of KMnO_4,_ the solution became brown. Next, that solution was diluted with water and then treated with hydrogen peroxide. Lastly, the product was washed with distilled water and 10% HCl solution to remove impurities. An improved form of the Hummers’ method for the preparation of GO was reported, by improving the oxidation with the addition of extra KMnO_4_ without NaNO_3_ addition, and a reaction was carried out in H_2_SO_4_/H_3_PO_4_ with a 9:1 ratio. This improved form of the Hummers’ method showed an even carbon network, more oxidized hydrophilic carbon, and no toxic gas production during preparation [39]. The phase purity and functional groups were initially confirmed by X-ray diffraction (XRD) and Fourier transform infrared spectroscopy (FT-IR). The surface morphology and microstructure of GO were confirmed by scanning electron microscopy (SEM) and transmission electron microscopy (TEM). By using Raman spectroscopy various graphene-based nanomaterials were characterized. Moreover, X-ray photoelectron spectroscopy (XPS), thermo-gravimetric analysis (TGA), differential scanning calorimeter (DSG), and atomic force microscopy (AFM) were used to evaluate graphene-based nanomaterials [40].

Graphene and GO have more surface area and strong light absorption properties, hence being considered ideal applicants in cancer therapy. Moreover, graphene has been confirmed to possess better photothermal anticancer efficiency than carbon nanotubes. The authors also concluded easy preparation, low cost, and low toxicity made graphene-based nanomaterials an ideal candidate for cancer treatment [34]. A later work evaluating the cytotoxicity of GO and GO loaded with doxorubicin (DOX) on human multiple myeloma cells suggested low-cytotoxicity GO as a suitable nanocarrier for anticancer drug [41]. Moreover, further work to improve GO biocompatibility was carried out by its initial conjugation with NH_2_-PEG3500-maleimide. Then, functionalization was performed using peptide (integrin αvβ6-specific HK) through maleimide-thiol coupling, and finally, HPPH was loaded on GO-PEG-HK via π−π stacking (Figure 1A). GO(HPPH)-PEG-HK was capable of killing the tumor cells and lung metastasis [42].

A novel mechanochemical method was developed to synthesize GO-Fe_3_O_4_ nanocomposites [45]. An efficient, nontoxic PEGylated GO/epirubicin was designed to destruct tumor cells [46]. In addition, hypocrellin A (HA) was loaded onto GO for anticancer treatment. The carboxyl, hydroxyl, and epoxide groups present on GO were linked with the quinone portion of HA via hydrogen bonding, as shown in Figure 1B [43]. Moreover, a PAH/FA/PEG/GO siRNA complex for gene delivery consistin of a GO monolayer delivering HDAC1 and K-Ras siRNAs to target pancreatic cancer cells was reported. The detailed synthesis procedure for PAH/FA/PEG/GO siRNA is shown in Figure 1C [44]. The combined use of PEG and grafted GO (pGO) enhanced its aqueous stability followed by loading of pGO with chlorin e6 (Ce6) photosensitizer and doxorubicin (DOX). Higher photodynamic anticancer effects as compared with Ce6/pGO or DOX/pGO were found [47]. Additionally, a covalently bonded biocompatible GO-PEG showed toxicity for lung cancer A549 and human breast cancer MCF-7 cells. Further, paclitaxel (PTX) was conjugated with GO-PEG via π-π stacking and hydrophobic interactions, and the results showed high toxicity to A549 and MCF-7 cells [48]. For the cancer cell apoptosis, a multifunctional FePt-DMSA/GO-PEG-FA (iron platinum-dimercaptosuccinnic acid/PEGylated GO-folic acid) composite was reported [49]. For the breast cancer cells, a PEGylated nGO loaded with PS and two-photon (TP) compound was prepared. The results showed that GO-PEG (TP) has the capability to kill breast cancer cells (4T1) at a 980 nm laser irradiation [50].

For bacterial infection phototherapy, a variety of metals and metal oxides were loaded onto graphene as antibacterial agents. Briefly, the TiO_2_-Ag/graphene as a ternary nanocomposite was synthesized and its photodynamic effect was carried on *E. coli* bacteria and A375 (melanoma), HaCaT (keratinocyte) cells. The results suggested the ternary composite could be applied for bacterial keratosis or skin tumors [51]. Additionally, different metals such as Zn, Ni, Sn, and steel were coated with GO (Figure 2A). The different metals have different capacities to fight against bacteria: GO-Zn acts as a better antibacterial agent than GO-Ni, followed by GO-Sn and GO-steel [52]. Moreover, a ZnO/GO nanocomposite prepared by loading green-synthesized ZnO NPs to GO nanosheets (Hummers’ method) (Figure 2B) has the capacity to kill the bacteria and also serves as an anticancer drug [53].

Additional advantages of graphene include (1) cross-linked capability with polymers, (2) admirable biocompatibility in in vitro and in vivo, and (3) more surface area, specifically graphene sheets. Hence it makes more contact with the bacteria and leads to more pronounced antibacterial effect. Considering all these advantages, a boronic acid-functionalized graphene and combined with quaternary ammonium salt (B-CG-QAS) acted as a multidrug-resistant to bacterial infection [54]. In addition, GO-PEI-GQDs via layer-by-layer deposition [55] and polyvinyl-N-carbazole-GO (PVK-GO) nanocomposite [56] were also reported. The variation in antibacterial effect due to variation in the combination of the substrate with GO was noticed. The GO fixed titanium with enhanced photoacoustic performance (GO-EPD) showed enhanced antibacterial, activity followed by GO-APS (GO-electrostatic interaction) and GO-D (GO-gravitational effect) [57].

### 2.2. Reduced Graphene Oxide

The RGO has been used in drug delivery, bioimaging, and anticancer applications due to its high electrical and thermal conductivity. However, GO has less NIR absorption capacity than RGO. Moreover, RGO is superior for high photothermal conversion and optical properties. The hydrophilic nature of RGO is essential in medical applications; hence, several efforts have been researched to enhance its hydrophilicity [58]. Furthermore, RGO was synthesized by chemical or thermal reduction of GO or graphite oxide. The hydrazine, hydrazine hydrate, sodium borohydride, and L-ascorbic acid are used as reducing agents during RGO synthesis [59]. Moreover, the plant extract is also used for the synthesis of RGO due to its non-toxicity, cost-effectiveness, biocompatibility, and environment-friendly nature over chemical and physical approaches (see Figure 2). As per the report, these biomolecules, such as amino acids, bovine serum albumin, humanin, glucose, melatonin, and ascorbic acid, interact with functional groups present in RGO [60]. Furthermore, humanin has been used for the green synthesis of RGO [61]. Chitosan was used to combine with RGO to reduce and stabilize the GO as well as entrap DOX and IR820 dye. The in vitro and in vivo results confirmed the chit-RGO-DOX-IR820 was applicable for cancer theranostics [62].

Various studies have reported the increased effectiveness of cancer treatment on combination therapies. For instance, GO (from graphene flakes) was partially reduced with NaOH and chloroacetic acid followed by surface modification to form FP-PrGO-Ce6-AuNR by depositing gold nanorods (AuNR) onto FP-PrGO-Ce6. The FP-PrGO-Ce6-AuNR nanocarrier acted as a targeting agent for anticancer theranostics [63]. Moreover, RGO-coated polydopamine doped mesoporous silica was used for anticancer treatment. The RGO/MSN/PDA-loaded DOX helps photothermal activity and shows an antitumor effect [64]. A green approach used an environmentally friendly, non-toxic, natural phenolic resveratrol compound instead of hydrazine and hydrogen sulfide for the formation of RGO [65]. Recently, an HSA/RGO/Cladophora glomerata bio-nano composite was prepared as a PS for study using L929, HeLa cancer cell line, Pseudomonas aeruginosa, and Staphylococcus aureus bacteria. The results showed that the synthesized composite has the capacity to kill bacteria with demonstrated photothermal activity [66].

For bacterial infection problems, silver is a well-known antibacterial agent. Moreover, combination therapies showed increased antibacterial properties. In addition, RGO induces photothermal effect for bacterial treatment. For instance, RGO/Ag composite was prepared as an antibacterial agent [67]. Moreover, the RGO-Cu_2_O nanocomposite was synthesized to fight against bacteria [68]. The GO/nitrogen-doped carbon dots/hydroxyapatite/titanium film (GO/NCD/Hap/Ti) showed a PTT/PDT approach to bacterial infection [69].

### 2.3. Functionalization

The synthesis of stable and functional GNCs is the most crucial aspect of the biomedical field. Though GO and RGO are reported as good PTT agents, its NIR absorption capability has to be improved further for more efficient phototherapy results. Moreover, RGO is hydrophobic, which limits its application during cancer treatment [37]. Hence, to fulfil these drawbacks, surface functionalization is the best option in the medical field to treat cancerous cells. Generally, the surface functionalization was carried out by covalent and non-covalent interactions. Non-covalent bonding includes electrostatic interactions, hydrogen bonding, π-π stacking, and van der Waals interactions. For example, a MFG (magnetic and fluorescent graphene)-SiNc_4_ (silicon napthalocyanine bis(trihexylsilyloxide) via covalent and non-covalent π-π stacking was prepared. MFG showed flat NIR absorption and was reported to have a remarkable PTT/PDT in HeLa cancer cells [70]. GO quantum dots (GOQDs) using hypocretin A (HA) for loading via π-π interaction were applied to detect cancer cells [71]. In addition, nucleophilic substitution, condensation, and electrophilic addition offer alternative paths for the covalent functionalization of GO.

Alternative approaches, such as GO with polyamidoamine dendrimer (GO-PAMAM) loaded with DOX and MMP-9 shRNA plasmid (Figure 3A), were applied for the treatment of breast cancer cells [72]. Moreover, the NGO-COOH prepared using Hummers’ method was functionalized with a Gd-DTPA dendrimer and finally loaded with the anticancer drugs epirubicin (EPI) and Let-7g miRNA (Figure 3B) to treat cancer cells [73]. Moreover, the GO-PLL(poly-L-lysine)/DOX/ZnPc acts as an admirable anticancer carrier to transport DOX and ZnPc to detect cancer cells. The synthesized nanocomplex shows not only anticancer activity but also photodynamic and chemotherapeutic effects against cancer cells [74]. Moreover, the GO firstly composited with carboxymethyl chitosan (CMC) followed by conjugation with hyaluronic acid (HA) and fluorescein isothiocyanate (FI) to prepare GO-CMC-FI-HA/DOX. The results proved that the nanocomplex can be used as an anticancer drug with controlled release [75].

The increase in bacterial infections is a serious problem for human health. Hence, the synthesis of multifunctional materials would be beneficial for surgical operations. Concerning this situation, GO functionalized (noncovalent) PEGylated phthalocyanines were synthesized for antibacterial phototherapy (ZnPc-TEGMME@GO) (Figure 4A) [76]. Moreover, RGO was functionalized with polycationic poly-L-lysine (PLL) to increase its drug loading capability with colloidal stability, as shown in Figure 4B. Further, RGO-PLL was labelled with anti-HER2 to form a bond with HER2 receptors to detect breast cancer cells [77]. The GO/AuNRs was synthesized and functionalized with polystyrene sulfonate (PSS) which showed tumor-killing capacity [78]. Furthermore, functionalization of RGO with hyaluronic acid increases the stability and cytocompatibility, as well as induce cancer cell ablation [79]. Moreover, the ZnO QDs-GO nanocomposite was prepared as an antibacterial agent. The author has combined chitosan with ZnOQDs@GO to enhance drug delivery capacity, biodegradability, and biocompatibility [80].

## 3. Graphene Nanocomposite Theranostics for Multimodal Imaging Guided Phototherapy

Nanomaterials-based theranostics are the future of personalized medicine as a single nanoplatform can provide multiple imaging and therapies in a short time by simplifying the cost and the amount of the drug required for multiple diseases [81]. Multimodal imaging-based diagnosis is the most reliable technique to identify the problems in cancer- and bacteria-infected patients, and it will be helpful to surgeons and clinicians to make better predictions and conclusions about the problem, thus will improve treatment confidence. Several imaging techniques have been adopted in the research, such as CLSM, MRI, CT, PET, PAI, Raman, ToF-SIMS, and other imaging techniques for early diagnosis (Figure 3) [82,83,84,85].

Phototherapy (PT) involves light interaction (Vis-NIR) with nanomaterials to generate heat or reactive oxygen to destruct cancer and bacterial infection. If the therapy process involves generation of heat from nanomaterials which can suppress or burn the tumor/bacteria is called PTT. If the PT involves reactive oxygen species (ROS) and singlet oxygen (^1^O_2_) generation to destruct the cellular components, it is called PDT. If the nanomaterial inherently cannot generate ROS and ^1^O_2_, it has to functionalize with PS [70]. The NIR light has high tissue penetration and low absorption by the biological medium. Hence, we usually adopt the NIR lasers for PT. Apart from PT, combination therapies with chemo and gene therapies could also enhance the treatment results [86]. Various PT agents have been explored by researchers including inorganic, organic, and composite nanomaterials [24]. The extensive publication record of functionalized nanographene composites on imaging guide therapy shows they are a focus in this field of theranostics due to their effectiveness in disease eradication. To overcome the individual drawbacks of diagnosis and therapy, integration of independent techniques has become a major challenge in nanomedicine.

### 3.1. CLSM for Imaging Guided Therapy

Typically, scientists rely on CLSM to identify cell morphology and drug internalizations as they are economic and the most available handy preliminary techniques in the lab. When the nanodrug is added to the cells before going to the CLSM, optical microscopes-based imaging is highly important to check the cell structure and morphology. After that, CLSM is helpful to identify the fluorescent drug molecule’s internalization, its location in the cells, and whether it was entered into the cytoplasm and thereby nucleolus or was hindered at the cell wall.

Quantum dots (QDs)-based imaging has drawn the attention of nanomedicine scientists due to its tunable size and variable colors with stable fluorescence emissions [87]. On the other hand, its toxicity issues lead to a focus on alternate materials. In this perspective, Au nanomaterials are said to be a hallmark for imaging guided therapy due to their tunable size and stable emissions with good biocompatibility [88,89]. However, carbon-based nanomaterials, such as graphene and CQDs, are emerging and attractive nanomaterials due to their high surface area and versatile surface chemistry to functionalize inorganic and organic imaging and drug molecules for multimodal imaging-guided therapy for cancer and bacterial infection. They are also reported to be highly biocompatible and antibacterial [90]. In order to be an imaging probe, graphene must be functionalized with luminescent inorganic or organic materials. In pioneering works by Dai et al. on the preparation of nanoGO-based imaging probes for imaging and therapy of cancer, GO was functionalized with PEG- and B-cell-specific Rituxan antibody for targeted cell imaging and cancer therapy [91,92]. Later, GO was functionalized with PEG and fluorescein to make the GO as highly biocompatible and fluorescent to monitor its internalization into the cells. Few more GO-based fluorescence imaging probes have been successfully reported [93,94].

We prepared a multifunctional graphene (MFG) by functionalization with polyacrylic acid, FeNPs, and fluorescein ortho-methacrylate to impart both magnetic and fluorescence properties for CLSM imaging in HeLa cells and in zebrafish whole-body imaging (Figure 5). The MFG showed good biodispersability, biocompatability, and stable green fluorescence emission inside the biological system. These results confirmed that the MFG could be a good candidate for imaging guided PTT of cancer and bacterial infection as it also possessed flat absorption in the entire VU-Vis-NIR region [95,96]. In order to make it a PDT drug, we functionalized a PS (SiNc_4_) to offer MFG-SiNC_4_ and the synergistic effects of PTT/PDT and PTT, as shown in Figure 5C,D, with 98% efficacy in light [70]. Very recently, mesoporous silica (MS) coated RGO was synthesized and functionalized with indocyanine green (ICG), PEG (MS-RGO-ICG-PEG), and folic acid (MS-RGO-ICG-PEG-FA) for targeted imaging and phototherapy of cancer. Figure 5G shows the increase in the temperature with laser irradiation at the tumor site of mice injected with MS-RGO-ICG-PEG-FA. Hence, it could be a good candidate for phototherapy, which was evident after the experiments. Figure 5H marks that there is a decrease in the tumor volume of the mice treated with MS-RGO-ICG-PEG-FA compared to MS-RGO-FA alone, without PS ICG. The effectiveness of tumor suppression can be explained by the synergistic effects of PTT from rGO and PDT from ICG. The experiments are performed using an 808 nm laser with 1 W/cm^2^ for 10 min [97]. Irrespective of this progress, fluorescence-based techniques are still limited to the laboratory stage presumably due to the limitations of poor resolution, and because the drug molecules should be fluorescent.

### 3.2. MRI for Imaging Guided Therapy

Among all imaging techniques, MRI and CT scans are clinically versatile and best used so far for imaging-based diagnosis purpose. These techniques do not rely on any fictionalization of fluorophores and QDs. MRI works based on the radio waves and magnetic field to identify damaged (cancer) tissue from the healthy tissues based on activating the local proton environment. It has the great advantage of ease of use for X-ray imaging techniques. However, it has the limitations of poor sensitivity and lengthy signal recording times. The proton magnetic moment of tissue is environment-dependent and the T1 and T2 times may not produce a better image, hence some external contrast agents are frequently used. The well-known Gd^3+^ for bright contrast and superparamagnetic iron oxide NPs for dark contrast are used [98].

Graphene-related nanomaterial magnetic composites have a great advantage in helping these imaging guided therapy processes to improve the signals or contrasts to provide enhanced resolution in the final images during disease identification [99]. GO has been a scaffold for many imaging probes and drug loadings, and there are several works that have discussed magnetic nanoparticles-loaded GO, creating a better contrast agent due to its high loading capacity [100]. Recently, the oxidation of ball-milled graphite producing a nanoGO was reported. The extent of oxidation along with the presence of Mn^2+^ ions from KMnO_4_ are responsible for better proton relaxivity and displayed very good T1- and T2-weighted MRI contrast images [101]. Moreover, the RGO and created structural defects and oxygen functionalities are reported too. The destruction of symmetry in RGOs sublattices created a paramagnetic property, and it was demonstrated to be a good MRI contrast agent. The authors have suggested that the amount of the defects and the oxygen functionality determines the paramagnetism [102]. A nonmagnetic particles-based GO by the fuctionalization of GO with fluorine for MRI with NIR absorption capability for photo therapy of cancer was reported [103]. Apart from this, metal-free, magnetic graphene QDs doped with boron provided very good MR imaging results in both in vitro and in vivo [104]. Moreover, a GO-DTP-Gd magnetic complex for T1 MRI was prepared and demonstrated to be a better contrast agent than commercially used Magnevist. The complex has further functionalized with doxorubicin through physorption and shows very good toxicity towards cancer [105]. In addition, graphene encapsulated cupper probes were prepared and used as neural electrodes to image neural-cell activities in the brain [106]. Further, ^99m^Tc^I^ and Gd-based pegylated ultrasmall nano GO (^99m^Tc^−^ and Gd-usNGO-PEG) were prepared for the multimodal MRI and SPECT/CT imaging of lymph nodes. The preparation approach is claimed to be chelator free, and the final product has been utilized for multimodal purposes [107].

In brief, many authors have reported that GO- and RGO-based iron oxide nanocomposites are excellent for MR imaging and guided therapy of cancer [100]. As discussed earlier, we also prepared an MFG-SiNC_4_ with excellent superperamagnetic properties, fluorescence, biocompatibility, and water dispersability for in vitro MRI to serve as a good contrast agent, as shown in Figure 6A,B. Later, we demonstrated this material for guided PT of cancer (Figure 5C,D) [70,95]. Further, polyethylene glycol and super paramagnetic iron oxide nanoparticles functionalized rGO (rGO-IONPs-PEG) was prepared for multimodal imaging, such as the MRI-, CLSM-, and PAI-guided PTT of breast cancer in both in vitro and in vivo. The prepared GNC had excellent magnetic properties for MRI imaging, with good fluorescence and PAI imaging capabilities and good drug loading capability. Figure 6C,D show the PT efficiency of RGO-IONPs-PEG-Laser and greater tumor reduction was observed after the laser ablation compared to controls. Figure 6E shows the guided MRI images of mice before and after injection, and during therapy, with and without lasers. The day 7 laser-treated MRI reveals the complete tumor absence compared to untreated mice at the same duration of time. The study [108] is among the papers which have demonstrated imaging-guided therapy with reliable MR imaging in a systematic manner for theranostics, as shown in Figure 6.

Apart from these, many researchers have synthesized GO/RGO based magnetic nanocomposite for MR imaging [109,110,111]. In addition to GO and rGO based IONPs, Gd doped graphene QDs also has been prepared and demonstrated for MR imaging [112,113], which could also offer good loading of imaging and drugs molecules and less toxicity to serve as a theranostic material. According to the literature the contrast agents improve the image quality and here the IONPs produce better contrast than Gd or other magnetic nanomateial composites.

### 3.3. CT for Imaging Guided Therapy

CT scan is an alternative 3D imaging technique whch works based on the X-ray attenuation of molecular tissues to identify tumors at lower cost than an MRI. It can also provide good spatial–temporal resolution of cancer tissues, and the image recording time is shorter than MRI. Usually, barium sulphate and iodine-based contrast agents are used when CT lacks the necessary sensitivity to image some soft, low-density tissues. The literature on various nanomaterials for CT scan is available. Carbon-based nanomaterials have a unique role due to their biocompatibility and high contrast agent’s immobilization capability. Among the few nanomaterials reported for CT scanning, Bi nanomaterial is exclusively studied, though CuS and some of its composites are also explored for CT and MR imaging together [98].

However, the available literature is limited regarding GNCs for CT scanning along with MR imaging. GNCs are synthesized from the oxidation of graphite with KMnO_4_ and reduced with HCl before being further functionalized with iodine, and have shown good CT and MR imaging contrast [114]. It has been reported that Bi NPs functionalized graphene QDs was prepared with good dispersibility and low toxicity for improved CT imaging followed by PT of cancer [115]. GO functionalized with FePt NPs composite was made and successfully demonstrated for both MR and CT imaging followed by in situ pH responsive targeted cancer inactivation [116]. A nanocomposite of BaHoF_5_ decorated GO-PEG was prepared with good biocompatibility, and was well demonstrated as a CT and MR imaging agent followed by PTT therapy [117]. GO decorated with ultra-small ZnFe_2_O_4_ and upconversion luminescence nanoparticles (UCNPs) have been well demonstrated for CT scanning along with MRI, PAI, and fluorescence imaging guided phototherapy as a unique multi diagnosis platform [118].

Recently, GO decorated with AuNPs, SPIONPs, along with DOX-loaded 1-tetradecanol (TD) was prepared (called smart nanocomposite, NC) and successfully demonstrated for CT- and MRI-guided controlled chemo-phototherapy of cancer in vitro and in vivo. The advantage of this material is that the Au NPs on GO can act as a CT contrast agent and provide better emission of X-rays due to its high atomic number (Figure 7A) with good house-filed values (Figure 7B). The FeNPs can act as dual CT and MRI contrast agents (Figure 7C,D). Hence, the combination of these two materials on top of GO has provided a great advantage of dual modal imaging and revealed increased contrast in concentration, after 24 h, at the post-intratumoral site (i.t) compared to the post intraperitonial site in the images of Figure 7C,D. Figure 7E shows a reduction in tumor volume within a few days after PT with DOX-NCs+NIR. Figure 7F shows the corresponding morphology of tumor and total mice view from pre injection to 90 days of PT [119].

Apart from these three (CLSM, MRI, and CT) well used techniques PET, PAI, and ToF-SIMS, and combined MRI/CT, CT/PEI, MRI/CT/PL/PAI, and MRI/CT/EPR, could also be important tools to the earlier diagnosis. These are currently of great interest to nanomedicine researchers.

### 3.4. PET for Imaging Guided Therapy

Radio labelling techniques such as PET and CT have great sensitivity due to the low background, requirement of low signal amplification, and good penetration depth in vivo. As said earlier, nanographene-based scaffolds have a greater importance in radio imaging technology with the functionalization of radioactive elements such as ^198,199^Au, ^64^Cu, ^66^Ga, ^111^In, and ^121^I. In this regard of PET imaging, for the first time GO-based targeting and non-targeting imaging agents such as ^64^Cu-NOTA functionalized GO-PEG (^64^Cu-NOTA-GO) and ^64^Cu-NOTA-GO conjugated with TRC105 (^64^Cu-NOTA-GO-TRC105) for targeting a CD105 (endogline) has been synthesized. ^64^Cu was linked with GO-PEG via 1,4,7-triazacyclo nonane-1,4,7 triacetic acid (NOTA, a chelating agent of ^64^Cu). As shown in Figure 8A–D, the ^64^Cu-NOTA-GO-TRC105 has very good biodistribution and targeting ability towards 4T1 tumor-bearing mice compared to non-targeting ^64^Cu-NOTA-GO, pre injected TRC105 blocking dosed mice, and CD105-negative MCF7 human breast cancer cells. The combined CT and PET images also can be seen in the images [120].

After that ^66^Ga was functionalized to GO and obtained the same result with similar strategy of PET imaging of 4T1 cells in vitro, in vivo and ex vivo with good distribution in the body without toxicity [121]. Later, the same researchers labelled the ^64^Cu to the rGO with the same synthetic adaption to prepare ^64^Cu-NOTA-rGO-TRC105 for targeting 4T1 murine breast cancer cells and obtained successful results than with non-targeting rGO nanodrug [122]. Very recently, radioactive iodine (^124^I)-labelled GO nanocomposite has been reported for PET imaging and boron delivery inside mice. The images reveal that time-dependent biodistribution in the liver, spleen, stomach, and heart for long-time circulation inside the body, up to 48 h. The in vitro studies of *C. elegans* confirmed that the ^124^I-GO does not show any significant toxicity. Hence ^124^I-GO could be a better candidate for boron neutron capture therapy of cancer [123].

### 3.5. PAI for Imaging Guided Therapy

PAI is another non-invasive imaging model to monitor the tumor environment with greater resolution and high tissue penetration depth. The PA signal production involves the following process. The light energy absorbed by the material converts into heat and increases the temperature, followed by thermoelastic expansion which causes the generation of acoustic waves (AWs). The AW generates an image contrast respective to the concentration of the absorbing material. In this regard, light absorbing nanomaterials have an advantage of converting photo energy into thermal energy and generation of AWs for better PAI imaging [124,125]. Among carbon nanomaterials, graphene-based 2D nanomaterials and their composites draw great interest in the study of PAI imaging due to their unique light absorption from UV, VIS, to NIR-I and NIR-II regions. Based on the belief of the light absorption of graphene, RGO are highly advantageous than GO as the later has a poor absorption of light in the visible and NIR regions [126]. Graphene-based nanoplatelets and nanoribbions (GNRs) are tested for PAI and thermal acoustic imaging (TAI) imaging. The oxidized GNRs were found to reveal dual modal PAI and TAI imaging [127]. Similar results of NIR absorption of microwaves reducing RGO-based PAI has been reported, and it is believed that the imaging intensities are wavelength-independent [128]. In the same year, dye-enhanced NIR absorption of GO was reported to overcome the limitations of GO absorption in NIR region to produce PA images for phototherapy of cancer. It was observed that the GO-ICG-FA (indocynine green an NIR absorbing dye and folic acid, a tumor targeting agent) showed better contrast, and no contrast was observed with GO-ICG or GO-FA alone [129]. Graphene microbubbles as an enhanced NIR PAI contrast agent was also reported with good biocompatibility and spatial resolution [130]. Another work describes that GO functionalized with chitosan–FA (GO-CS-FA) has good success as a PA and fluorescence tumor vascular imaging guided therapy for cancer in vivo [131].

A dual modal PAI and photothermal imaging probe rGADA nanocomposite was fabricated by the rGO functionalized with AuNS (gold nanostars), bilayered lipids, FA (rGADA), and K-Ras gene plasmid (KrasI) rGADA-KrasI, for targeted imaging guided photothermal and gene therapy of pancreatic cancer. The Figure 9A shows that there is a good PAI contrast with increasing concentrations of rGADA. The Figure 9B reveals in vivo tumor PAI imaging at different times from 0–48 h showing that a distinct rGADA at tumor has been apparent with time. Figure 9C,D are photothermal curves for the temperature rise of the nanocomposite and photothermal images at 808 nm. From the information obtained from the above experiments it is evident that the rGADA has successfully internalized and distributed at the tumor site for PTT and gene transfection of cancer in vivo. In vivo PTT experiments showed 76.1% tumor suppression under laser with rGADA+L, whereas the gene therapy results in 55.2% with rGADA-KrasI. The combined PTT and gene therapies ofrGADA-KrasI + L with laser resulted in very good tumor suppression of 98.5% compared individual therapy and therapy without a laser. The measured comparative weights are shown in Figure 9E of the tumors for controls and rGADA-KrasI after laser irradiation. They demonstrate that a negligible and completely vanished tumor was evidenced [132]. Similarly, rGO–AuNPs also reported to PAI for NIR–II phototherapy [133]. Hence, graphene-based targeted multiple imaging guided combination therapies could be a very good idea in the implementation of non-invasive theranostic probes.

### 3.6. Raman for Imaging Guided Therapy

Raman is a spectroscopic technique named by its inventor Sir. C. V. Raman in 1928 who proposed the Raman Effect. The phenomenon is based on the inelastic and elastic scattering of light from the vibrations of objects such as nanomaterials, drugs, and other biological molecules. When this technique is coupled with microscopy it is called Raman microscopy, and can be used to visualize in vitro and in vivo biological components, internalized nanomaterials, and drugs with high specificity and sensitivity at workable spatial resolutions. However, high sample concentration is required for better resolution and fast image acquisition due to the weak scattering signals. To overcome this hurdle, Resonance Raman (RR), Surface-Enhanced Raman Spectroscopy (SERS), and Coherent Raman spectroscopy (CRS) were developed. Scientists have mostly adopted SERS in nanomedicine due to its signal enhancement in the presence of rough surface nanomaterials such as Au, Ag, and Cu in the sample system [134].

A GO@Au and fluorescent tag functionalized dual modal luminescent and Raman imaging has been reported [135]. Moreover, the GO-Ag nanocomposite for a SERS-based imaging cellular probe and FA for targeting the tumor to impart was prepared. The GO-Ag-FA treated cells have shown excellent uptake and cellular internalization and evidenced by SERS images taken after 2 h of incubation time [136]. In addition, a AuNR@GO nanocomposite functionalized with DOX to obtain DOX@GO@AuNRs for chemo and PT of HeLa cancer cells was reported. The GO and AuNRs showed strong SERS signals, but the DOX signals decreased within the cells due to the phagocytosis and the acidic environment inside the cells. The prepared nanomaterial showed good SERS signals and temperature changes upon laser irradiation. Hence it demonstrated good chemo-PTT results under light and was titled as a two-step Raman guided therapy [137]. After GO, an RGO-based, SERS-guided, low laser-powered, targeted PT was reported by preparing anti-EGFR-PEG-rGO@CPSS-Au-R6G. The RGO was functionalized with PEG, CPSS (carbon porous silica nanosheets), Au nanosheets, R6G, (Rhodamine 6G a Raman reporter), and anti-EGFR (epidermal growth factor receptor for targeting tumor) for sensitive low-powered laser-efficient NIR PT therapy against A549 and MRC-5 cells [138].

Recently, a SERS-guided multi modal chemo, gene, and PT of cancer with Au@GO-NP-NACs was reported, where the NP stands for nanoparticles and NACs for nucleic acid components ex. BCL2 mRNA. Figure 10A shows the schematic image of the guided SERS imaging and therapy in vivo. There is a tumor microenvironment which depicts the heterogeneity of the tumor, and the laser illumination of the tumor and normal tissue projects the effectiveness of the tumor eradication with Au@GO-NP-NACs. Figure 10B is for schematic representation of SERS signals at non-tumor and tumor site and the SERS intensity mapping, where the intensity of Cy5 is high at non-tumor tissues and less at the tumor. Figure 10C shows the corresponding Raman spectrum intensities at 1120 cm of Raman dye in different tissues. From the observation of the mapping and the spectral intensity, the SERS was varied, and we observed very weak signal at tumor in vivo due to the over expression of BCL2 in mice after the injection of the graphene drug. This kind of analysis is indispensable to evaluate drug distribution and its circulation in healthy and unhealthy tissues for specific and effective eradication of cancer. After the therapy process, the tumor from tissue has removed and observed its volume compared with untreated tumors. It was found that there is a great reduction in its volume in (Au@GO-NP-NACs) NP-NIR-treated mice, as shown in Figure 10D. The comparative tumor volume and time of therapy with control, NIR, and non-NIR treated NPs (Au@GO-NP-NACs) was plotted and it was apparent that the therapy was highly effective with NIR laser and NPs after 3 weeks (Figure 10E). The same material was functionalized with DOX and other types of genetic materials to evaluate the combined chemo-gene–PT of cancer to provide a better outlook of the therapy results in a single and minimal dosage of a drug, within a short time, with non-invasive NIR lasers. Such efforts for evaluating the potential of single material theranostic ability are highly warranted in nanomedicine to clear the hurdles of clinical trials [139].

### 3.7. ToF-SIMS for Cellular Imaging and Guided Cancer Therapy

ToF-SIMS imaging is one of the most surface-sensitive techniques to analyze chemical compositions of materials and biological systems containing chemical components. It has great capability to map low molecular weight components (<500 KD) and submicron resolution. ToF-SIMS involves sputtering the primary ion beam (Bi_3_^+^, Ar_n_^+^, and C_60_^+^) with the sample surface, and the secondary ions generated from the sample will be collected according to their flight times and its mass/charge. The chemical compositions can be predicted based on their respective masses in comparison with the reference library. This technique is unique in imaging single cells, human tissues, and skin and cancer cells, and could be an important label-free tool to diagnosis the cancer cells from healthy normal cells. It is helpful in studying the toxicity of nanomaterials, drug internalization into the cells, apoptosis, and to predict other cellular killing mechanisms by collecting and imaging the cellular components’ mass/time values [140,141]. The information obtained also has great importance in predicting the drug and cellular interactions, hence also in drug development and pharmacology studies [142]. However, it has a limitation of low sensitivity in analyzing the very low molecular weight components in wet samples, as the large molecules’ excessive fragmentation obtained with a high energy ion beam is not very accurate. As a result, any material which could enhance the signal sensitivity has priority, as every single fragment is indispensable in predicting the disease information. GO and graphene have been used as a matrix material for enhancing surface sensitivity and signal intensity in analyzing small lipid molecules [143].

The potential toxicity by ZnO NPs in sun cream is of increasing concern. We have developed ToF-SIMS and CLSM imaging methods using human skin equivalent HaCaT cells as a model system for rapid and sensitive ZnO NPs cytotoxicity study (Figure 11A). The CLSM images (Figure 11B) revealed the absorption and localization of ZnO NPs in the cytoplasm and nuclei. The TOF-SIMS images demonstrated elevated levels of intracellular ZnO concentration and associated Zn concentration-dependent 40Ca/39K ratio, presumably caused by the dissolution behavior of ZnO NPs (Figure 11C). The imaging results demonstrated spatially-resolved cytotoxicity relationship between intracellular ZnO NPs, ^40^Ca/^39^K ratio, phosphocholine fragments, and glutathione fragments [144].

In a recent study, the ToF-SIMS signal enhancement of the single layer graphene covered wet cells with Bi_3_^+^ as a primary ion source was reported. The secondary ion imaging of cholesterol at *m/z* 369.25, phosphoethanolamine at *m/z* 142.05, palmitic acid at *m/z* 255.25, and oleic acid at *m/z* 281.26 are mapped [145]. An earlier study on the signal enhancement of ToF-SIMS by amine functionalized graphene quantum dots (GQDs) also show a better signal enhancement compared to hydroxyl GQDs in a comparative study [146]. From the above discussion it was evident that, GO, GQDs, and graphene have a remarkable effect on the quality of ToF-SIMS spectra and imaging and can overcome the hurdles of wet cell imaging’s complex matrix effects. Non-invasive multimodal imaging by a single nanoprobe could offer a greater advantage of gathering the diagnosis information from each technique by providing its advantage where an individual imaging technique cannot. It will improve diagnosis accuracy and efficiency. Moreover, the multiple nanoprobe-based nanotheranostic material offers minimal toxicity and provides body–blood clearance easily by avoiding multiple drug dosages. In brief, each technique has its advantages and disadvantages. However, highly sensitive, non-invasive techniques could take a greater importance than other techniques in nanomedicine in the future.

### 3.8. Guided Phototherapy of Bacteria

Bacterial infections have led to millions of patients dying every year all over the world. Generally, antibiotic treatment has been used for bacterial infections. However, inappropriate and overuse of antibiotics has led to an increase in the drug-fighting capacity of bacteria [146]. Notably, antibiotic resistance is related to structure transformation, gene mutation, and bacterial biofilm formation. Additionally, biofilm is a multicellular bacterial group surrounded by its own synthesized extracellular polymeric material composed of proteins, polysaccharides, lipids, and extracellular DNA [147]. The extracellular polymeric material provides an appropriate microenvironment for bacterial growth, and protection against antibiotics, and hence, bacterial infection control becomes an obstinate challenge. Thus, there is an urgent need to find new strategies to combat bacterial infections [148]. PTT gained increasing demand in the medical field over conventional antibiotic therapy because it destructs bacteria and their biofilm. Specifically, PTT combined with NIR light has various benefits, including deep tissue penetration, spatiotemporal controllability, and little light absorption in tissue. Nevertheless, the disadvantage of PTT is that a nonselective thermal effect may arise due to the weak affinity between pathogenic bacteria and a photothermal agent that may damage healthy cells during irradiation [149].

Increasing bacterial infections are a serious problem for human health. Hence, the synthesis of multifunctional antibacterial materials is needed for surgical operations. Concerning this situation, we prepared non-targeted and targeted magnetic graphene and carbon nanotubes against *S. aureus* and *E. coli* for PTT. Excellent bacterial capturing efficiency (Figure 12A,B) was observed with MRGOGA (magnetic RGO functionalized with glutaraldehyde). This was also evident from the SEM images shown in Figure 12C. The batch-mode- and continuous-mode PTT showed 99% killing efficiency under NIR laser irradiation at 808 nm, shown in Figure 12B. The plate count method, shown in Figure 12D, demonstrated that both the strains had completely vanished after laser treatment with MRGOGA [150,151].

We have started working on phototherapy of bacterial infection with progressive achievements firstly using ZnO NPs [152] followed by using modified carbon nanotubes [153] and lastly extends to biomimetic applications using graphene nanomaterials [154,155,156,157]. The trend of using graphene nanomaterials is just at the beginning. For instance, the preparation of GO-functionalized (noncovalent) PEGylated phthalocyanines was used for antibacterial phototherapy (ZnPc-TEGMME@GO). The antibacterial activity against *E. coli* and *S. aureus* bacteria at different illumination was shown in Figure 13A,B. As reported, the synthesized nanocomposite showed PTT/PDT capacity with antibacterial activity. The authors further recorded SEM images before and after the treatment of nanocomposites against bacteria. The formation of holes on the bacterial surface confirmed the damage to the cell membrane. Further, the material was demonstrated in vivo by considering mice as a model animal. From the thermographic images it was confirmed that the material was internalized and can create local heating around the wound. Hence, it favors the in vivo PT/PDT, and the results after irradiation with 450 nm (PTT) and 680 nm (PDT) confirmed complete wound healing after 12 days of treatment. Whereas the control mice has the persistent wound even after laser irradiation without photodrug (Figure 13C–E) [76]. In addition, the concept of targeted nanoparticles in cancer therapy with in vivo biocompatibility of graphene-based nanomaterials is summarized. The detailed chemistry and properties of GO as well as the review of functionalized GO and GO-metal nanoparticle composites in nanomedicine for anticancer drug delivery and cancer treatment is reviewed [158]. Moreover, the concept of targeted nanoparticles in cancer therapy with in vivo biocompatibility of graphene-based nanomaterials is summarized. The detailed chemistry and properties of GO as well as the functionalized GO and GO-metal nanoparticle composites in nanomedicine for anticancer drug delivery and cancer treatment is reviewed [159].

In another study, the RGO was functionalized with polycationic poly-L-lysine (PLL) because of more drug loading capability with colloidal stability. Further, rGO-PLL is labeled with anti-HER2 to form a bond with HER2 receptors to detect breast cancer cells [159].

### 3.9. Comparison among GNCs

According to the above research discussion, every author made their contribution towards this field. Dai et al., for the first time, introduced the GO and RGO to the theranostic applications, thereby extensively contributing to this important research. Later, other researchers gave their insights to fuel the graphene nanomedicine. For instance, we have prepared MFG to impart long-range absorption of graphene in both biological window I and II, along with demonstrating the CLSM/MRI and both single light induced PTT/PDT [70,95,96]. Choi et al., Yang et al., Mirrahimi et al., Hong et al., Jia et al., Yang et al., Belu et al., and Lim et al. demonstrated multimodal imaging (CLSM, MRI, CT, PET, PAI, RAMAN and ToF-SIMS) and multimodal therapies including chemo, gene, and PTs. Most of these researchers have also given tremendous efforts to improve the therapeutic capabilities by minimizing GNCs concentration, laser wavelengths from the first biological windows to the second biological window, and less laser powers and irradiation times [84,97,107,108,119,120,132,139,145,146]. Keshav et al., provided excellent pharmacokinetic data to take the material towards preclinical trials. Table 1 represents some of the interesting works discussed. Based on the comparison of the tabulated literature, 808 nm laser with irradiation time of 5–15 min with ~1 W/cm^2^ and ~100 µg/mL were the most suitable parameters for phototheranostics. In the case of bacteria, the same parameters are good for photodisinfection. However, according to us and to Liang et al., experiments with very low powers and GNCs dosage have also shown great therapeutic effects [96,150,158]. As each datum is indispensable, we appreciate the existing literature greatly. Apart from GNCs, there are several nanomaterials which have been reported for nanomedicine. Among the reported carbon, Au, Fe, Si, dendrimer, and polymer nanomaterials are highly suitable as novel theranostic agents [160,161,162,163,164]. Graphene has a great advantage over other nanomaterials due its tunable size, dimensionality, tunable surface, covalent and non-covalent chemistry, atomic sensitivity and <nm thickness, easy synthesis, and its economic availability for both cancer therapy and antibacterial activity [165,166,167].

## 4. Conclusions and Future Perspectives

We summarized the recent progress of the general preparation and functionalization of GO, RGO, and GNCs as theranostic materials to provide simple and advanced imaging-guided therapeutic drugs to invade malignant tumors and bacterial infections. The water solubility, low toxicity, and high surface area of GO made a very good nanoplatform to carry many therapeutic organic drugs and to load different imaging probes. However, its low NIR absorption is unlikely, and not very favorable to the phototherapy of cancer and bacteria. Hence, RGO or functionalized nanocomposites of graphene-related materials provide a better solution to overcome the difficulties where GO cannot. The multi-modal imaging and PS functionalized nanographene composite provide a very accurate diagnostic confidence to proceed with the therapy of combined PTT/PDT, which may require in less time and smaller drug concentrations. Among the nanotherapies reported, phototherapy has good results, with less intensive time and energy, and without any side effects and damage to healthy tissues.

Graphene/GO/GQDs can offer diversified chemistry for self-acting luminescent for CLSM, magnetic for MRI, surface plasmonic state for SERS and ToF-SIMS signal enhancement, PAI imaging, and inherent PTT, PDT agent. It has great potential to carry many chemical drugs and genes for chemo- and gene therapies with very good biocompatibility. However, much research is required to move GNCs towards clinical implementation, as their size, shape, no of carbons, layers, number of oxygen functional groups, accurate mass, and photo yield to generate ROS and heat have to be optimized precisely. In perspective of PT, the biological windows must be explored in NIR-I and NIR-II. Overall, nanotechnology scientists could use flexible GNCs in whatever they want to fabricate.

## Data Availability

Not applicable.

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
