# Peer review of "Multimodal Imaging and Phototherapy of Cancer and Bacterial Infection by Graphene and Related Nanocomposites"

_molecules, 2022, doi:10.3390/molecules27175588_

Round 1

Reviewer 1 Report

In this review, authors reported cancer and infectious diseases by the graphene and their nanocomposite. Author first showed the general Introduction of the important case of graphene in the multimodal imaging guided therapy. And then the part of preparation of graphene nanocomposites are detailed showed the various methods in the synthesis and functionalization of the graphene and their nanocomposite. Specially, authors also showed application of GO in the cancer treatment. Finally, the conclusions and future perspectives also presented in the end. In total, this manuscript is logic and complete in length. However, this is an old topic of GO and related materials, and I already see many published papers there. Authors should highlight your own characteristics to meet the requirement of the Molecules.

1.     Authors must provide at least two tables to compare Go and other materials for drug delivery. One is compared with other materials such as polymers, silica, metal, etc. another should self-compared to show the different design in the Go for targeted applications.

2.     To be a beautiful review, the size and front should be similar in the Figures, the fonts and images in some of the figures are obviously not clearly visible. Such as Figure 11a, Figure 12ab. Also, sometimes, Arial, sometimes Time new Roman.

3.     What is the benefit of the GO materials compared to other materials? It should compare and discuss it. Some related research about the drug delivery materials should be cited and compared with Go. Advanced functional materials, 2020, 30(2): 1902634. Nano Res. 15, 5556–5568 (2022). https://doi.org/10.1007/s12274-022-4160-6. Biomolecules, 2022, 12(5): 636. Biomater. Sci., 2022, DOI: 10.1039/D2BM00719C.  J Control Release. 2022 Aug 6:S0168-3659(22)00485-0. doi: 10.1016/j.jconrel.2022.08.005.

4. How about the application of Go-based nanocarriers overcome drug resistance?

5. How is the clinical translation of the Go materials and the Go-based nanocarriers? How many products are under clinical translation?

Reviewer 2 Report

Here in this review, the authors have summarized the recent progress of general preparation and functionalization of GO, RGO, GNCs as theranostic materials to provide simple and advanced imaging guided therapeutic drugs to invade malignant tumours and bacterial infections. In perspective of PT, the biological windows have to explore completely in NIR-I and NIR-II. Over all GNCs could state as the rational choice of the nanotechnology scientists to be flexible in whatever they want to fabricate. I believe that publication of the manuscript may be considered only after the following issues have been resolved.

1.      For the abbreviation appearing for the first time in the full text, the author needs to make some remarks, such as, WHO, SMIS, CLSM, MRI, CT, PET, ToF-SIMS, and so on.

2.      Some pictures are not clear enough, and the text information in the pictures is extremely unclear. The author needs to make adjustments, such as, Figure 2, Figure 3, Figure 4, Figure 5, Figure 11, Figure 12.

3.      There are too many keywords in the article, and the author needs to simplify them.

4.      Since this review not only introduces graphene, it is suggested that the author adjust the title as follows: Multimodal Imaging and Phototherapy of Cancer and Bacterial Infection by Graphene and related Nanocomposites.

5.      In order to increase the readability of the paper, in the introduction, the author needs to mention some work on graphene and related composite materials, such as RSC Adv., 2022, 12(13), 7821-7829; RSC Adv. 9(2019) 41383–41391; Electrochim. Acta, 168(2015) 337–345; Talanta, 134(2015) 435–442; RSC Adv., 8(2018) 42233–42245.

6.      In scheme 1 and scheme 2, the author should provide some corresponding references.

Reviewer 3 Report

This review article provide a detailed summary  regarding the progress towards developing graphene-based nano-composite for multimodal imaging and phototheraphy .

Overall, the manuscript is in good standard and well detailed. Therefore, I recommend this to accept with a minor revision.

(1). Can authors briefly explain the advantages of nano composites over small molecule probes or fluorescence proteins?

(2). Conclusion section may need a minor modification to cover a complete summary of the manuscript.

(3). There are minor grammatical and textual errors in the manuscript.

Round 2

Reviewer 1 Report

I'm glad the author solved my problem better

Reviewer 2 Report

Accept in present form.